# The Application of Cone Beam Computed Tomography (CBCT) on the Diagnosis and Management of Maxillofacial Trauma

**DOI:** 10.3390/diagnostics14040373

**Published:** 2024-02-08

**Authors:** Arif Rashid, Lee Feinberg, Kathleen Fan

**Affiliations:** 1St George’s University Hospitals NHS Foundation Trust, London SW17 0QT, UK; arif.rashid@nhs.net; 2King’s College Hospital NHS Foundation Trust, King’s College London, London SE5 9RS, UK; lee.feinberg1@nhs.net

**Keywords:** cone beam computed tomography, CBCT, maxillofacial trauma

## Abstract

The assessment and management of facial trauma in an acute setting is one of the core services provided by oral and maxillofacial units in the United Kingdom. Imaging is a pre-requisite for appropriate diagnosis and treatment planning, with a combination of plain radiographs and medical-grade CT being the mainstay. However, the emergence of cone beam CT in recent years has led to its wider applications, including facial trauma assessment. It can offer multi-planar reformats and three-dimensional reconstruction at a much lower radiation dose and financial cost than conventional CT. The purpose of this review is to appraise its potential indications in all anatomical areas of maxillofacial trauma and provide our experience at a level 1 trauma centre.

## 1. Introduction

Maxillofacial injuries are common, accounting for 5–10% of emergency department attendances in the United Kingdom (UK) [1]. Injuries can range from an isolated fracture to complex comminuted panfacial fractures, often with concomitant injuries in the polytrauma patient. Radiographic examination forms an integral part of the diagnosis, surgical planning, and outcome assessment of these patients. In the diagnostic setting, imaging should depict the location, extent, and degree of displacement of fractures and soft tissue damage. Traditionally, the standard of care was a combination of plain films perpendicular to one another. However, with the increasing availability of cross-sectional imaging, conventional helical computed tomography has superseded plain films, particularly in the assessment of upper and mid-third facial fractures. 

Cone beam computed tomography scanners (CBCT) started development in the late 1990s for use in dentistry. Like conventional CT, CBCT can offer a three-dimensional (3D) view that plain films fail to provide. It uses a low-energy fixed anode tube, similar to that used in a dental panoramic radiograph machine, that rotates around the patient once, capturing data using a pyramidal-shaped X-ray beam. The field of view (FOV) can be adjusted to capture only the area of interest, allowing for the radiation dose to be ‘as low as diagnostically acceptable’ [2]. The result is a high-resolution, real-size dataset allowing for multi-planar reformats and 3D reconstruction, ideal for fracture assessment [3]. Images are of higher geometric accuracy and spatial resolution, with less scatter when compared with conventional CT [4], derived from a smaller and cheaper machine, and with reduced radiation [5,6].

CBCT is now widely used in the UK for risk assessment in lower third molar surgery. Its increased accessibility to maxillofacial units has lent itself to wider applications, namely for diagnosis and outcome assessments in maxillofacial trauma. The purpose of this article is to appraise the literature on its use in the trauma setting and to provide the author’s experience of CBCT as an imaging adjunct at a tertiary referral trauma centre. We present the evidence base to support the wider utilisation of CBCT in facial trauma assessment, highlighting its specific indications for each anatomical region and its increasing use in the intraoperative setting.

## 2. Cone Beam CT Dosimetry and Safety in Facial Trauma

X-rays are the source of radiation used in both CT and CBCT imaging. X-rays are a type of electromagnetic radiation towards the higher frequency end of the electromagnetic spectrum. They are a type of ionising radiation, meaning that atoms become ionised when they are exposed to this radiation. This can result in subsequent damage to the DNA molecules within human cells, and the main risk in the context of diagnostic imaging is malignancy developing in the patient. Although the risks from these imaging modalities are very low, there is no safe dose, and as the dose increases, the risk increases.

The International Commission on Radiological Protection (ICRP) defines key principles of radiation protection that are carried forward into local legislation in different countries; justification and optimisation are two of these principles [7]. Justification is the idea that a patient should only be exposed to ionizing radiation if the benefit outweighs the risk of harm. The Royal College of Radiologists (RCR) states that ‘a useful investigation is one in which the result—positive or negative—will inform clinical management and/or add confidence to the clinician’s diagnosis’ [8]. Optimization is the idea that every reasonable attempt is made to reduce unnecessary doses. Doses of ionizing radiation should be ‘as low as reasonably practicable’ (ALARP), with economic and social factors being considered [7]. Based on these principles, it would seem reasonable to perform three-dimensional imaging if this is likely to add confidence to the diagnosis and help ensure that the most appropriate management or surgical approach is utilised. Given that CBCT and conventional CT can both provide high-resolution imaging of the hard tissues, it seems sensible to consider the use of CBCT as a potentially lower-dose alternative in order to comply with the ALARP principle.

There are different ways in which doses can be presented. The most basic being the absorbed dose, which is measured in Grays. This simpler measure of dose considers the amount of radiation energy deposited per unit mass of tissue. This measure of dose can be readily established and can be taken as a readout from a scanner. However, it does not consider how ionising that type of radiation is, and it does not consider the radiosensitivity of the tissues. The gold standard measure of dose is the effective dose, as this considers all the above factors. This is measured in Sieverts. Given that the effective dose considers the radiosensitivity of the tissues irradiated, effective doses from imaging of different body parts can be directly compared. The effective dose can also be directly correlated with the risk of cancer induction. The risk of cancer induction from dental radiography is 1 in 15,000,000/μSv for men and 1 in 18,000,000/μSv for women [9]. Unfortunately, effective doses are more difficult to calculate and therefore are not available as a readout from a scanner, and in our case, we have obtained data from academic publications. 

Doses for both CT and CBCT will be variable depending on the equipment used and exposure factors selected. However, a meta-analysis of published data, which included nine CBCT units, showed an average effective dose (μSv) for an adult to be 212 µSv for large FOVs. A large field of view CBCT is classified as a volume greater than 15 cm in height, therefore imaging the full maxillofacial skeleton [10]. Whereas when imaging a similar field of view with CT, an effective dose of 860 µSv was produced in another study by the same author [11]. In a further study, a range of 685 µSv to 1410 µSv was shown for CT imaging [12]. Therefore, CT imaging is, in many cases, four or five times the dose of CBCT imaging of the same anatomy. 

In our institution, full-face imaging can be obtained with CBCT for a dose of around 80 µSv [13], which is potentially one tenth of the dose of CT. However, there must still be a justification for using CBCT imaging, as doses are still higher compared to conventional 2D radiography. The dose of a panoramic radiograph is around 20 µSv [14], and the dose of a single facial X-ray is around 10 µSv [15]. Therefore, a typical series of 2D imaging for facial trauma is 20–30 µSv (panoramic radiograph + PA mandible or OM 0 + OM 30 views), compared to around 200 µSv for CBCT imaging and 1000 µSv for CT imaging (Table 1). Therefore, although CBCT imaging offers a five-time dose advantage compared to CT, it is still around eight times the dose of 2D imaging. In many cases, this dose is justified by the significant increase in diagnosis confidence and detail for surgical planning that comes from 3D imaging. 

Table showing the relative doses of imaging modalities in facial trauma.

## 3. Applications for Cone Beam CT Imaging in Maxillofacial Trauma

Hereafter, we describe the applications for CBCT in the assessment of trauma in the different anatomical areas of the face. A summative table of the key primary studies is provided in Appendix A. We also provide a concise guideline outlining the principles for selecting imaging examination methods for each region of the maxillofacial skeleton in Table 2.

The table details the author’s recommendation for imaging modality choices for fracture assessment for each anatomical region of the face.

### 3.1. Cone Beam CT for Assessment of Dentoalveolar Fractures

Dentoalveolar injuries are a common presentation to maxillofacial units, with a global prevalence of 15.2% in the adult dentition, 22.7% in the primary dentition, and a median age of 13.8 years [16]. Low-velocity injuries are more likely to cause damage to the supporting structures, resulting in luxation and avulsion injuries. Conversely, high-velocity injuries deliver higher energy to the crown of the tooth and are therefore more likely to produce crown and root fractures [17]. Occlusal and periapical plain films are the standard radiographic examinations utilised to assess these injuries. However, CBCT has been shown to be more sensitive when assessing for root and alveolar fractures and resorption [18,19]. The ability of a plain X-ray to detect a root fracture depends on the angle of the X-ray beam and the degree of separation of the fractured segments. One study demonstrated a detection rate of 30–40% with periapical radiographs, compared with a 90% detection rate with CBCT [20]. However, both imaging modalities demonstrated limitations when detecting vertical root fractures [21]. CBCT may also be better at assessing for pulp and periodontal healing [22,23] and thereby have a place for follow-up assessments. 

Figure 1 is a small-volume CBCT scan of the anterior maxilla showing luxation injuries of the UR1 and UR2 with an associated alveolar bone fracture that extends to the anterior nasal spine. This cross-sectional imaging allows for a more detailed assessment of the supporting structures of the teeth, which may influence management decisions. Figure 2 is an example of root fractures demonstrated on CBCT imaging that were occult on plain films. 

It is the author’s opinion that plain films and clinical examination should remain the standard of care for initial assessment, especially as this management is time-dependent. Exceptions would include high-velocity injuries, which benefit from assessment with cross-sectional imaging. CBCT with a small FOV should be considered in these cases, especially if there is a suspected root fracture or concomitant alveolar bone fracture, in which the approach to management may be modified.

### 3.2. Cone Beam CT for Assessment of Mandibular Fractures

The mandible is fractured in up to three-quarters of patients with maxillofacial fractures [24]. In the demographic area served by the authors unit, interpersonal violence is the dominant cause, with the angle and parasymphysis the most commonly fractured sites, followed by the mandibular condyle [25]. Clinical examination coupled with plain radiographs (orthopantogram and posterior/anterior mandible X-ray) is often adequate to arrive at a diagnosis. The presence of a malocclusion, sublingual haematoma, step deformity, and inferior dental nerve paraesthesia can be readily assessed. Moreover, Markowitz et al. [26] found no statistically significant difference between the sensitivities for mandible fracture identification on CT compared with plain films. 

Others argue that CT offers imaging-enhancing tools, improved image quality, and decreased interpretation error [27], overcoming the problems with superimposition and anatomical structure distortion in the anterior mandible and coronoid/condylar region innate to plain films. Kaeppler et al. [28] in a prospective study reported that an additional 17.75% (*n* = 41) occult fracture sites and 14.72% (*n* = 34) additional infractures were identified on CBCT (medium or large FOV was used to image both condyles), not noted on clinical examination and plain radiographs. Further information about displacement (23.8%, *n* = 55) and comminution (3.46%, *n* = 8) were also provided. In their study, this additional information, particularly occult fractures, resulted in a change in treatment in 9.52% (*n* = 22) of cases, which was statistically significant. Incomplete cortical plate fractures and their extension to the dentoalveolar segment can be better appreciated. 

Condyle fractures are classified according to location (condylar head, condylar neck, subcondylar, and dicapitular) or direction of fracture (horizontal, vertical, and sagittal) [29]. CBCT offers better visualisation of TMJ anatomy, degree of fracture displacement, and comminution, which may influence surgical management. To this end, the American Academy of Oral and Maxillofacial Radiology published a position paper in 1997 [30], recommending cross-sectional imaging only in complex condylar fractures. Raustia et al. [31] found that both the antero-posterior and medio-lateral displacement of the fracture were better visualised on cross-sectional imaging. Similarly, Orhan et al. [32] when assessing vertical condylar fractures created on cadaver specimens, found CBCT was far more sensitive. In cases of intracapsular fractures, some argue the benefit of magnetic resonance imaging (MRI) to assess for internal derangement that can be simultaneously treated. Wang et al. [33] demonstrated on MRI that 91.5% of condyle fractures with dislocation showed anteroinferior disc dislocation, 87.3% showed retrodiscal tissue abnormal signal intensity, and 85.6% showed joint effusion.

It is the author’s opinion that plain radiographs are the gold standard of imaging assessment for simple mandible fractures. There is a lack of robust research to support the additive use of CBCT. However, CBCT may be considered for comminuted and segmental fractures, as the ability to reformat and 3D print the images will aid surgical planning and pre-operative adaptation of plates. It also has its place in condylar fractures to better understand the direction of displacement and degree of comminution. CBCT is often used in the post-operative review setting, particularly when assessing for complications from metal work or inadequate bony healing. In a recent study [34], CBCT was used to predict post-operative inferior dental nerve injury, with associations including the proximity of the fracture line to the nerve canal and interruption along the course of the nerve. This could be of benefit to consenting patients, but further well-designed and large sample studies are needed before concluding on its predictive capacity.

Figure 3 shows plain radiographs and CBCT images of a patient presenting with an infection following open reduction and internal fixation of a left angle of mandible fracture. Cross-sectional imaging in this case was able to confirm the suspicion of non-union on the plain film, as well as the lingual bone loss and periapical area associated with the LL7 apices, suggesting a loss of vitality in that tooth. 

Figure 4 shows 2D images taken immediately post-operatively and CBCT images of the same patient at 24 months. The CBCT in this case can better visualise the position of the metalwork in relation to teeth and the inferior dental nerve. This added information can help inform decisions to remove metal work in patients with complications of nerve paraesthesia or non-vital teeth.

### 3.3. Cone Beam CT for Assessment of Frontal Sinus Fractures

Frontal sinus fractures account for between 5 and 15% of all facial fractures [35]. The most common aetiology is motor vehicle accidents. Fractures may involve the anterior or posterior table, or both, and may extend to involve the orbits, ethmoids, or nasal base [36]. They often occur with concomitant injuries, namely intracranial haemorrhage, cervical spine fractures, and other facial fractures [37,38]. For this reason, they are best investigated during the primary trauma survey with a conventional helical CT. CBCT does not provide adequate contrast resolution for assessment of underlying traumatic brain injury and therefore should not replace CT in such instances. Fractures at the level of the cribriform plate may be better visualised on CBCT, as unlike conventional CT, images are the same quality in all reformatted planes. For this reason, in a patient with persistent CSF rhinorrhoea, CBCT may be used to assess for occult fractures [39].

### 3.4. Cone Beam CT for Assessment of Orbital Fractures

The orbit is susceptible to injuries following blunt trauma owing to its fragile medial and inferior orbital walls. The AOCMF classification group categorises orbital fractures according to anatomical sub-regions: orbital rims, anterior orbital walls, mid-orbit, and apex [40]. Surgical exploration and reconstruction are warranted if there is muscle entrapment resulting in a functional impairment or if there are enophthalmos or hypoglobus with aesthetic implications. Non-contrast CT is the gold standard for imaging orbital injuries, providing multi-planar slices with good spatial resolution. Importantly, Ilanonkovan [41] found MRI more sensitive than CT for the diagnosis of herniation and entrapment of soft tissues, and thus MR may be utilised effectively if CT and clinical examination are not conclusive.

The lens is one of the most radiosensitive tissues in the body, and thus, radiation to the orbits will increase the risk of cataract development. CBCT generates fewer X-ray photons, resulting in less radiation at the expense of reduced contrast. Brisco et al. [42] compared the diagnostic quality of CBCT images of orbital fractures with conventional CT. They found the soft tissue contrast was poorer in CBCT, but despite this, the imaging of the extra-ocular muscles and optic nerve was close to that of conventional CT, and entrapment was appropriately detected. However, their sample size was small, and thus caution should be taken when drawing conclusions from their results. Moreover, a retro-orbital haemorrhage was depicted in one conventional CT, but not in the CBCT of the same patient. Similarly, Roman et al. [43] evaluated 93 CBCT images of midface trauma and noted a higher sensitivity in detecting orbital floor fractures when compared with CT sections with a high level of inter-examiner agreement. 

There is insufficient evidence to support the use of CBCT over conventional CT in the acute setting where there are concomitant facial or head injuries or where there is suspicion of muscle entrapment or retrobulbar haemorrhage. Figure 5 shows a CBCT coronal section of a patient with a right orbital floor fracture. The fat and muscle are indistinguishable when compared with the same slice in a conventional CT and therefore are not adequate for assessing muscle entrapment.

In the outpatient setting, in the absence of eye signs and a conventional CT, CBCT may be adequate for assessing the size of the defect and orbital volume in surgical planning. Furthermore, in this setting, mirror computational planning can be used and patient-specific implant (PSI) adapted free-hand on 3D-printed models. In the majority of cases, PSIs are acquired from conventional CT data. However, studies have shown the feasibility of using CBCT to construct orbital PSI [44]. At the time of publication, CBCT scans were not accepted for PEEKMilled implants, Titanium 3D-printed orbital and cranial implants by some medical technology companies (Synthes CMF CT-CBCT Scan Protocol) [45]. In our unit, CBCT is frequently utilised to assess the adequacy of orbital reconstruction (Figure 5).

### 3.5. Cone Beam CT for Assessment of Nasal Fractures

The nasal bones are the most commonly fractured facial bones [46], owing to their central position and prominent projection. A fractured nasal bone warranting intervention is usually apparent on clinical examination, and traditionally, imaging has not been added to diagnosis or therapeutic interventions. Indeed, a patient may have a displaced nasal bone fracture clinically, but normal-appearing occipto-mental X-rays [39,47] fail to demonstrate cartilaginous disruptions or fractures. 

CBCT can be employed if there is a suspicion of an associated fracture—nasomaxillary, naso-orbital-ethmoid, and floor of the frontal sinus. Coronal reformats can aid in understanding the fracture displacement and position of the septum in the absence of nasendoscopy and thereby guide instrument reduction.

### 3.6. Cone Beam CT for Assessment Fractures of the Zygomatic Maxillary Complex and Midface

The zygoma is the second-most commonly fractured facial fracture after nasal bone fractures. It contributes to the structure of the midface, articulating with several bones of the craniofacial skeleton, and thus fractures can be associated with significant functional and aesthetic morbidity. In the last decade, conventional CT has replaced occiptomental plain radiographs as the gold standard for imaging assessment. The ability to review imaging in multiple planes can aid the surgeon in understanding the direction of displacement to plan the reduction and fixation required for stability. Figure 6 demonstrates the difficulties with interpreting OM views owing to the superimposition of anatomical structures. The left zygomatic complex fracture is much better visualised on CBCT-reconstructed images.

When compared to conventional CT, a recent blinded and randomised study by Rozema et al. [48] assessed the reliability of CBCT with low-dose multi-detector CT on unilateral zygomaticomaxillary fractures on fresh frozen human cadaver specimens, demonstrating similar diagnostic capabilities. Heiland et al. [49] recommended CBCT over conventional CT in the absence of neurological symptoms or extensive injuries. 

Similarly, midface fractures comprising a Le Fort Pattern are best appreciated with cross-sectional imaging. These fractures often accompany a multi-injured patient, and thus conventional CT is more readily utilised in non-ambulatory and comorbid patients. Otherwise, CBCT can show a larger number of fracture lines and fragments [40], and 3D printing can facilitate surgical planning, the alignment of teeth, and the construction of custom arch bars without the need for impressions. In our unit, CBCT is often utilised to assess fracture reduction post fixation (Figure 7). This serves as a baseline post-operative image and a more reliable assessment of bone reduction than a plain X-ray.

## 4. Intraoperative Use of CBCT in Surgical Management of Facial Fractures

Intraoperative 3D imaging first developed prominence in orthopaedic surgery, with studies demonstrating better fracture reduction, implant positioning, and reduced need for revision surgery, particularly in cases of articular fractures or fractures in complex anatomical areas [50,51].

In the last decade, intraoperative 3D imaging has increasingly been used in maxillofacial trauma. Stanley [52] and Manson [53] published some early feasibility studies, noting that even among experienced surgeons, intraoperative clinical assessment of adequate reduction of zygomatic complex fractures or orbital reconstructions was often not confirmed by the post-operative computed tomographic (CT) scan. Intraoperative CT can allow for immediate evaluation of bony or implant positioning, thereby avoiding the risk of further surgical revision. 

Pohlenz et al. [54] described the first clinical applications of intraoperative CBCT with an integrated flat-panel detector in oral and maxillofacial surgery after surgical treatment of ZMC fractures. The C-arm CBCT is considered easy to use, with fewer space requirements compared with a medical-grade CT. Additional surgical time was estimated at 8 to 30 min, including setup, sterile draping, and image review [55,56,57]. The current CBCT C-arm has a limited FOV of 12 cm × 12 cm × 12 cm meaning the contralateral side cannot be imaged. Assouline et al. [55] suggest merging the intraoperative data set with pre-operative imaging to achieve this comparison.

Applications include the assessment of reductions in zygomaticomaxillary fractures (ZMC) and orbital reconstructions. Surgical approaches do not allow for direct visualisation, so ZMC reduction is conventionally assessed by sphenozygomatic suture reduction and subjective assessment of facial symmetry. In a retrospective study of 48 patients with zygoma or orbit fractures, intraoperative CBCT allowed for immediate revision in 6 of the 48 cases, particularly in cases of comminution [58]. Furthermore, in seven patients, the need for orbital exploration post-ZMC reduction was deemed no longer indicated, thereby preventing unnecessary orbital exploration. This is echoed by a study on intraoperative CT at a level 1 trauma Centre in Portland, Oregan [59], where CT-guided revision rates were reported as follows: orbital 31%, Zygomaticomaxillary complex 24%, Le fort I 8%, Le fort II and III 23%, naso-orbital ethmoid 23%, mandible 13%, and frontal sinus 0%. 

Other uses include assessing condyle reduction and screw fixation in relation to the inferior dental nerve [60] and intraoperative localisation of foreign bodies, particularly in the case of gunshot wound foreign bodies in close proximity to at-risk structures [61].

Although the rationale and benefit of intraoperative imaging can be appreciated, cumulative ionising radiation exposure from repeated CBCT imaging, especially if intraoperative adjustments have been made, is not without risk. Johner et al. [62] reported an average of 1.3 CBCT scans per patient, and this would correlate with the complexity of the trauma. Alasraj et al. [57] reported the highest number of revisions in ZMC cases (63.6%), and these were the only cases that required a second intra-operative scan. Moreover, despite the evidence supporting its use, a recent national French survey revealed that as little as 30% of university hospital departments and 0% of private clinics were using it, with the main indication for use being temporomandibular joint surgery or orbital fracture management [63].

## 5. Other Applications of CBCT in the Facial Trauma Setting

There are other applications for CBCT in the trauma setting. There is less scatter and artefact from metal compared with conventional CT, and therefore it can be used effectively when assessing facial trauma secondary to gunshot injuries [64] and for localising metal foreign bodies. It is superior to CT in detecting hard-tissue injury in the immediate locality of a high-density metal projectile. In addition, CBCT-guided removal of projectiles or metallic foreign bodies intraoperatively is associated with shorter operating times and fewer complications, including major bleeding, soft tissue infections, and nerve damage [61].

CBCT has a known application in the assessment of the airway in sleep-disordered breathing patients [65]. It can be used for geometric airway volume analysis, which can assist in anaesthetic planning (‘virtual laryngoscopy’) and endo-tracheal tube choice [66].

## 6. Limitations and Drawbacks of CBCT

The additive use of CBCT to a maxillofacial department will result in financial costs in terms of purchase, maintenance, and training. It offers lower contrast resolution when compared with conventional CT and cannot be used for measurement of Hounsfield units (HUs) [67,68]. There is an increased susceptibility to movement artefact, particularly in early-generation machines. Moreover, pathways and training in competent reporting of these images must be considered before implementing them as a new standard of care. 

## 7. Conclusions

Radiographic imaging of the trauma patient must comply with the ‘as low as diagnostically acceptable’ principle. The choice of imaging should be guided by the type and severity of the injury (Table 2). A combination of plain films will remain the initial screening assessment for low-level maxillofacial trauma. Further cross-sectional imaging with the ability to generate multi-plane reformats is of benefit in select mandibular trauma and the majority of midface and upper facial third trauma. In the absence of specific indications for a conventional CT, such as a concomitant head or C spine injury, CBCT offers an acceptable alternative with less radiation exposure and potential applications to all areas of the maxillofacial skeleton. Clinicians should understand these applications and their limitations. 

## Figures and Tables

**Figure 1 diagnostics-14-00373-f001:**
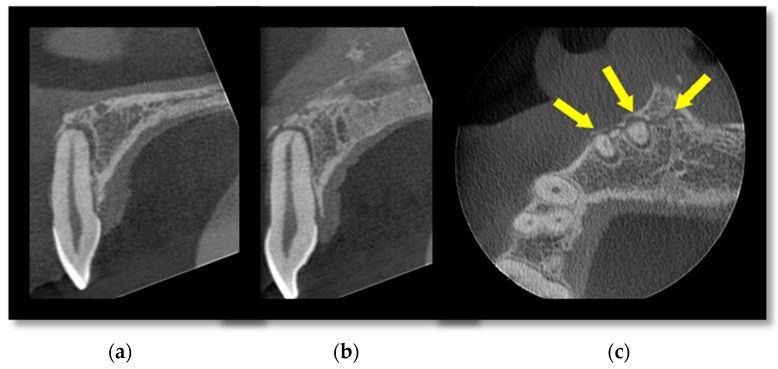
A small FOV CBCT of the anterior maxilla showing (**a**) luxation injury with (**b**) associated alveolar bone fracture (**c**) and extending to the anterior nasal spine demonstrated by the arrows.

**Figure 2 diagnostics-14-00373-f002:**
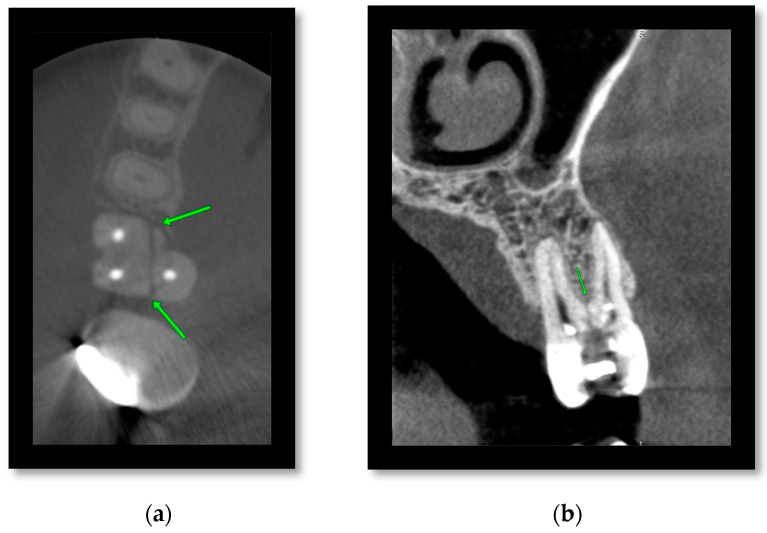
An (**a**) axial and (**b**) coronal section of a root-treated first maxillary molar tooth demonstrating a fracture through the furcation not evident on plain films. Arrows demonstrate the fracture.

**Figure 3 diagnostics-14-00373-f003:**
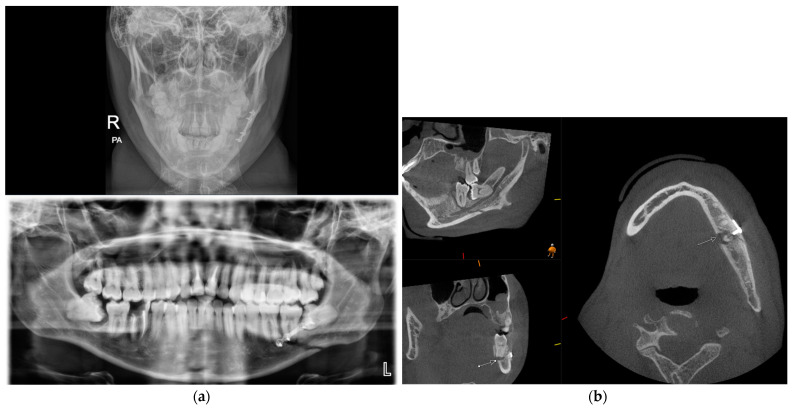
(**a**) PA mandible and OPG and (**b**) sagittal, coronal, and axial slices demonstrating non-union of a left angle of mandible fracture. CBCT better visualizes the area of non-union and the lingual bone loss around the LL7 tooth (see arrow).

**Figure 4 diagnostics-14-00373-f004:**
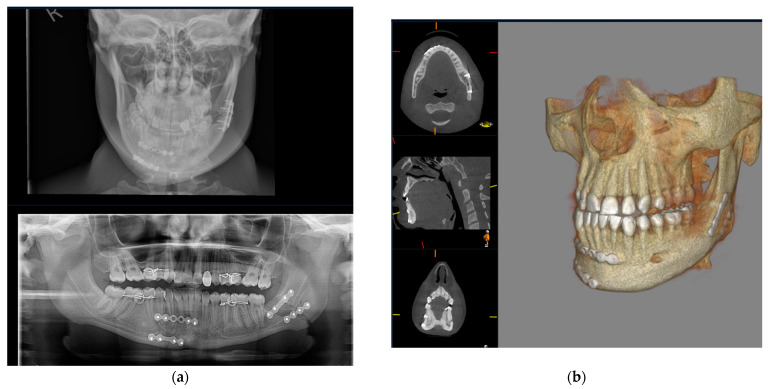
(**a**) PA mandible and OPG and (**b**) axial, sagittal, coronal reformats, and a 3D reconstructed image. CBCT can better visualize the position of metalwork in relation to teeth and the inferior dental nerve and thus help inform decisions on metalwork removal.

**Figure 5 diagnostics-14-00373-f005:**
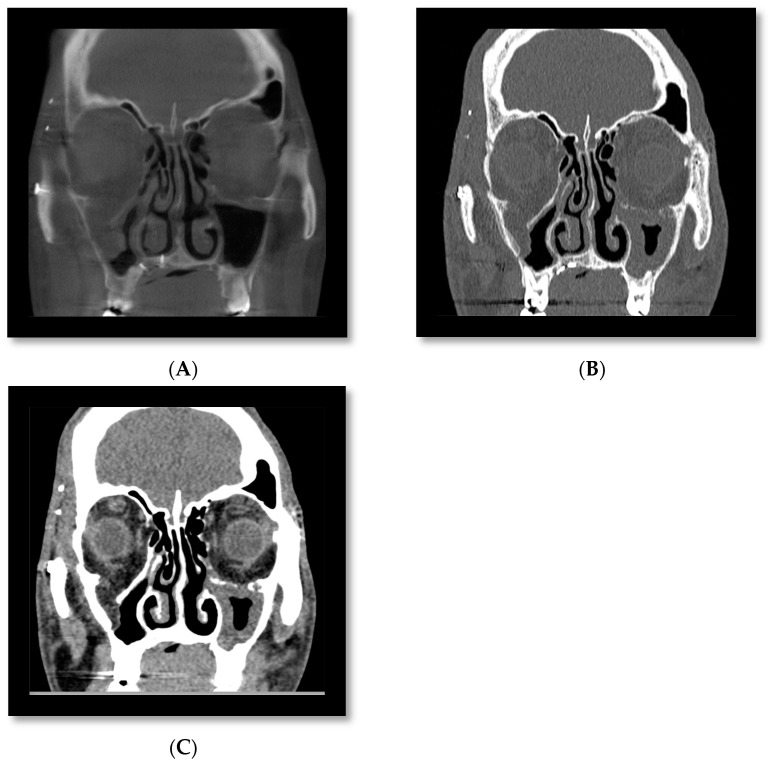
CBCT (**A**) CT bone window (**B**) and CT soft tissue window (**C**) of the same patient, showing a right orbital floor fracture with soft tissue herniation. CT soft tissue windowing allows for differentiation between fat and muscle. On CBCT fat and muscle are indistinguishable. CBCT is suitable for assessment of the bony injury, but it is not possible to assess for muscle entrapment on CBCT. CT (**D**) at time of acute orbital trauma shows a right orbital fracture involving the floor, medial wall, and orbital roof. Soft tissue windows show inferior and medial rectus herniation. CT (**E**) prior to revision orbital floor repair shows that the medial rectus is herniating into a persistent medial wall defect (see arrow). CBCT (**F**) post-revision orbital floor repair shows that the new custom plate is well positioned with repair of both the floor and medial wall.

**Figure 6 diagnostics-14-00373-f006:**
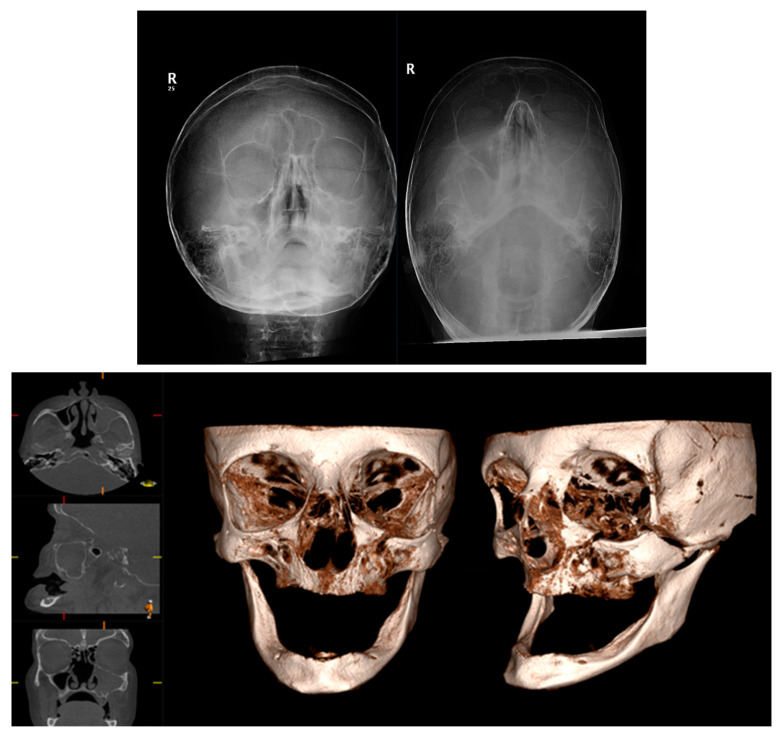
Occipitomental views and CBCT axial, sagittal, coronal, and 3D reformatted images demonstrating a fracture to the left zygomatic complex. Conventional occipitomental views are difficult to interpret owing to superimposition. CBCT-reformatted images show the comminuted nature of the fracture and also allow for visualization of the orbital floor.

**Figure 7 diagnostics-14-00373-f007:**
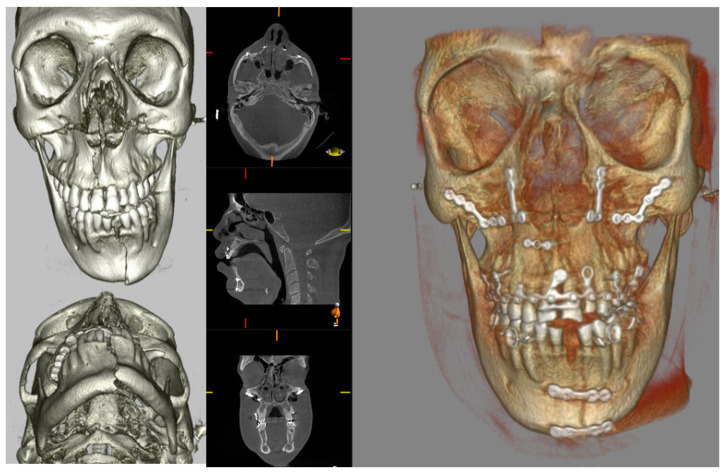
Pre-operative CT 3D volume renders and post-operative CBCT axial, sagittal, coronal slices, and 3D volume render showing reduction and fixation of fractures through the symphysis mandible, Le Fort 1 level, and maxillary alveolar bone.

**Table 1 diagnostics-14-00373-t001:** Effective Doses.

Imaging Modality	Effective Dose µSv
Facial X-ray	10
Panoramic X-ray	20
CBCT Full Facial Skeleton	200
CT Full Facial Skeleton	1000

**Table 2 diagnostics-14-00373-t002:** Choice of imaging according to anatomical area of interest.

Maxillofacial Injury Region	Suggested Imaging Modality
Dentoalveolar Injury	1st Line: Plain films (PA/occlusal/OPG) and clinical examination 2nd Line: CBCT if suspected occult root fracture or concomitant dentoalveolar fracture
Mandible Fracture	1st Line: Plain films (OPG/PA mandible views) 2nd Line: CBCT for comminuted and segmental fractures to allow for 3D printing and surgical planning; 3D assessment of condyle fractures; post-operative evaluation of complications with healing and metal work
Nasal Fracture	1st Line: Clinical examination 2nd Line: CBCT or CT if suspicion of naso-orbital-ethmoid fracture
Midface Fractures	1st Line: CBCT (if ambulatory and without suspicion of traumatic head injury and for intraoperative and post-operative assessment); plain radiographs (OM views) if isolated zygomatic arch fracture 2nd Line: Conventional CT if concomitant head injury
Orbital Fractures	1st Line: Conventional CT 2nd Line: MRI for detailed assessment of muscle entrapment; CBCT in the absence of eye signs for surgical planning and post-operative reconstruction assessment
Frontal Bone Fractures	Conventional CT for assessment of concomitant traumatic head injury

## Data Availability

No new data were created or analyzed in this study. Data sharing is not applicable to this article.

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
