# Peer review of "The Application of Cone Beam Computed Tomography (CBCT) on the Diagnosis and Management of Maxillofacial Trauma"

_diagnostics, 2024, doi:10.3390/diagnostics14040373_

Round 1

Reviewer 1 Report (New Reviewer)

Comments and Suggestions for Authors

Author Response

Thank you for your detailed review of this submission and your valued comments and acceptance without need for amendment. 

In response to your comment regarding references 1, 52 and 53 and their date of publication:

  1. Reference 1 dated 1998 detailing the incidence of maxillofacial trauma in UK emergency departments , has been updated to a reference from 2020.
  2. Reference 52 and 53 dated 1999 represent early feasibility studies into intraoperative CT scanning in maxillofacial trauma and therefore of interest in this publication. I have added to the main text that these are early feasibility studies.

I hope these amendments satisfy your comments.

Best Wishes

Mr Arif Rashid

Reviewer 2 Report (New Reviewer)

Comments and Suggestions for Authors

A well-structured review paper on an interesting topic. 

Author Response

Thank you for the positive feedback and for recommending this paper for publication without any further amendment.

Best Wishes

Mr Arif Rashid

Reviewer 3 Report (New Reviewer)

Comments and Suggestions for Authors

This manuscript primarily delves into the application of Cone Beam Computed Tomography (CBCT) in the assessment and management of facial trauma. The authors presented the indications for the use of CBCT in various anatomical regions of the oral and maxillofacial area, drawing from the author's wealth of experience. The piece is not only engaging but also comprehensive, offering a detailed discussion on the topic that will serve as a valuable reference for readers.

However, one notable caveat is that the author's proposed guidelines for the use of CBCT in the diagnosis and treatment of mandibular fractures lack robust research support. It would be beneficial for the author to clarify this limitation within the text.

Furthermore, to enhance the reader's understanding and ease of information absorption, it is recommended that the authors add a table outlining the principles for selecting examination methods for each anatomical region of the face. This addition would provide a concise and accessible overview, complementing the in-depth discussion presented in the article.

So, minor revision should be recommended for this manuscript.

Author Response

Thank you for reviewing this manuscript and for accepting the submission after minor amendment.

I have responded to your suggestions as follows:

1) 'the proposed guidelines for the use of CBCT in the diagnosis and treatment of mandibular fractures lacks robust research support'. I have added this limitation into the main text (lines 189-190).

2) ' Addition of a table outlining the principles for selecting examination methods for each region of the face'. I have added a summative table to serve as a guide for imaging selection choice based on the findings presented in this literature review. It is listed as Table 2 and can be found after the Conclusion section. 

I hope I have addressed the above comments adequately.

Best Wishes

Mr Arif Rashid

This manuscript is a resubmission of an earlier submission. The following is a list of the peer review reports and author responses from that submission.

Round 1

Reviewer 1 Report

Comments and Suggestions for Authors

I would like to thank the authors for submitting this manuscript. I enjoyed reading it very much. It is well written and nicely organized.

Reviewer 2 Report

Comments and Suggestions for Authors

Dear Authors,

I hope this message finds you well. After careful review and consideration of your submitted manuscript, I come to the conclusion that we will not be able to accept it for publication in our journal.

While we recognize the effort and time invested in compiling the information, the manuscript, as it stands, lacks the depth of scientific rigor that our journal seeks. The content primarily offers a descriptive overview of a technology that has been well-established and in use for at least the past 15 years. As such, it does not provide any novel insights or advancements that would be of interest to our readership.

Our aim is to publish articles that contribute new knowledge, innovative approaches, or fresh perspectives to the field. Unfortunately, the current submission does not meet these criteria.

We appreciate your interest in our journal and encourage you to continue your research endeavors. Should you have any future works that align more closely with our publication's objectives, we would be more than happy to consider them.

Thank you for your understanding, and we wish you the best in your future academic and research pursuits.

Warm regards,